# Hydroxyapatite Formation Coexists with Amyloid-like Self-Assembly of Human Amelogenin

**DOI:** 10.3390/ijms21082946

**Published:** 2020-04-22

**Authors:** Jing Zhang, Jian Wang, Chengwei Ma, Junxia Lu

**Affiliations:** 1School of Life Science and Technology, ShanghaiTech University, Shanghai 201210, China; zhangjing2@shanghaitech.edu.cn (J.Z.); wangjian1@shanghaitech.edu.cn (J.W.); chengwei.ma@ge.com (C.M.); 2State Key Laboratory of Molecular Biology, CAS Center for Excellence in Molecular Cell Science, Shanghai Institute of Biochemistry and Cell Biology, Chinese Academy of Sciences, Shanghai 200031, China; 3University of Chinese Academy of Sciences, Beijing 100049, China

**Keywords:** amelogenin, hydroxyapatite, enamel biomimetic, protein assembly structure, solid-state NMR

## Abstract

Tooth enamel is formed in an extracellular environment. Amelogenin, the major component in the protein matrix of tooth enamel during the developing stage, could assemble into high molecular weight structures, regulating enamel formation. However, the molecular structure of amelogenin protein assembly at the functional state is still elusive. In this work, we found that amelogenin is able to induce calcium phosphate minerals into hydroxyapatite (HAP) structure in vitro at pH 6.0. Assessed using X-ray diffraction (XRD) and ^31^P solid-state NMR (SSNMR) evidence, the formed HAP mimics natural enamel closely. The structure of amelogenin protein assembly coexisting with the HAP was also studied using atomic force microscopy (AFM), transmission electron microscopy (TEM) and XRD, indicating the β-amyloid structure of the protein. SSNMR was proven to be an important tool in detecting both the rigid and dynamic components of the protein assembly in the sample, and the core sequence ^18^EVLTPLKWYQSI^29^ was identified as the major segment contributing to the β-sheet secondary structure. Our research suggests an amyloid structure may be an important factor in controlling HAP formation at the right pH conditions with the help of other structural components in the protein assembly.

## 1. Introduction

Enamel, the material covering and protecting the teeth, is composed of hydroxyapatite (HAP) crystals [1]. Different from bone or dentin, the elongated hydroxyapatite crystals are interwoven into a highly organized microstructure in enamel, bestowing enamel with extraordinary strength and resistance to fracture [2]. Amelogenin, as the major component of enamel protein, plays a key role in regulating enamel mineral morphology [3]. Mutation of amelogenin causes amelogenesis imperfect disorders in human [4]. Knockout of amelogenin in mice results in defective enamel formation [5]. Although not fully understood, the biomimetic approach has utilized amelogenin to regenerate enamel-like materials on teeth in gentle and close to physiological conditions [6]. Full-length amelogenin [7], leucine-rich amelogenin peptide (LRAP) [8] and other amelogenin-derived peptides [9] have all been tested and proved to have great potential in dentistry or biomaterials. 

Amelogenin belongs to the group of intrinsically disordered proteins. In acidic solution, it is disordered [10]. In neutral or weak basic condition, it tends to oligomerize into various high-molecular weight structures, which depends on the solution environment [11,12,13]. During amelogenesis, ameloblast cells excrete the proteins, 90% of which are amelogenin and its variants [3]. The proteins form a matrix template guiding apatite crystal formation into elongated structures. During enamel maturation, the proteins are enzymatically removed to allow the crystal to expand in directions perpendicular to its long axis [2]. Many studies have been carried out to learn the protein assembly structures during amelogenesis. Transmission electron imaging techniques have identified amelogenin nanospheres in the enamel matrix [14], although structural information at the molecular level is still lacking. Utilizing X-ray diffraction, Jodaikin et al. revealed the molecular structure of developing rat tooth enamel at the molecular level [15]. They found 0.47 nm reflections, indicating some of the organic matrix proteins adopt a β-sheet conformation. Enamel matrices from the secretory stage and the maturation stage of kallikrein-4 knockout mice stained positively with Congo red, suggesting an amyloid structure with β-sheet conformation [16]. Solid-state NMR also revealed the β-sheet secondary structure for selectively ^13^C-labeled lysine residues of murine amelogenin at a functional state [17]. Therefore, a β-sheet conformation of amelogenin assembly has been proposed as the template to guide the mineral deposition and the enamel formation [18]. In vitro, amelogenin is able to form amyloid-like conformation in many conditions, from an oil-water emulsion system [19,20] to simulated enamel fluid [13], which closely mimic physiological conditions. However, no work has shown that amyloid-like conformation of amelogenin is functional in promoting enamel formation in a biomimetic environment. 

In the present work, we found that the human full-length amelogenin (H175) was able to form an amyloid-like structure at a pH range from 4.5 to 6.0. However, only at pH 6.0, which is in the pH range for natural enamel development [18], was amelogenin with this amyloid-like structure functional in promoting HAP [Ca_5_(PO_4_)_3_OH] formation. In contrast, the brushite (CaHPO_4_·2H_2_O) was the main mineral deposit at pH 5.0 and 5.5 conditions with H175 and pH 6.0 condition without H175. The structure of HAP formed at pH 6.0 in the presence of H175 was characterized using XRD and ^31^P solid-state NMR. Our results revealed that HAP formed by amelogenin in vitro closely resembles HAP crystals, using XRD, and human enamel, using ^31^P 1D and 2D ^1^H-^31^P heteronuclear correlation (HETCOR) spectra. Solid-state NMR (SSNMR) was also carried out to obtain structural details of H175 assemblies at pH 6.0 using uniformly [^13^C, ^15^N]-labeled protein, and the core sequence ^18^EVLTPLKWYQSI^29^ participating in the amyloid formation was identified. Besides this main β-sheet region, other rigid and dynamic parts in H175 are also discussed. 

## 2. Results

### 2.1. H175 could Adopt Amyloid-Like Structure at Acidic pH Conditions

Previously, we have showed that LRAP can assemble into an amyloid-like structure at acidic pH conditions [21]. In this work, human full-length amelogenin (H175) was tested for the amyloid-like assembly formation. In the presence of 33.4 mM CaCl_2_ and 20.9 mM KH_2_PO_4_, the protein aggregates at the screened pH conditions (4.5, 5.0, 5.5). At pH 4.5, the aggregation is gel-like and transparent, after leaving the sample at 37 °C for several days. At pH 5.0 and 5.5, the aggregation is white and forms quickly upon pH adjustment using KOH. The samples were usually collected after 2 weeks’ incubation or longer at 37 °C. The final mixtures were subjected to atomic force microscopy (AFM) examination (Figure 1a–c), showing smaller curvy aggregates at pH 4.5 and straight rod structures at pH 5.0 and 5.5. From the UV reading of the supernatant after ultracentrifugation of the mixture, we can conclude that about 30% of H175 is left in the solution for pH 4.5 and 5.0 and about 20% of H175 for pH 5.5 at the end of incubation. XRD of the total precipitates collected from ultracentrifugation clearly indicate two diffraction rings corresponding to the spacings of 4.8 Å and 10.1 Å for all three pH conditions (Figure 1d–f). The two distances are characteristic distances of the protein amyloid [22,23]. Thus, the results indicate that H175 would form amyloid-like assemblies at acidic pHs. 

### 2.2. HAP is Formed at pH 6.0 in the Presence of H175 Amyloid 

During enamel development and maturation, extracellular pH is rigorously regulated. It oscillates from neutral (~7.2) to weakly acidic (5.5–6.2) conditions [18]. In order to test whether amelogenin could adopt an amyloid structure at a natural pH condition, H175 was also incubated at pH 6.0 with 33.4 mM CaCl_2_ and 20.9 mM KH_2_PO_4_. At this condition, a large amount of white sand-like precipitates were observed immediately after pH adjustment, indicating mineral salt formation. Finally, about 20% of H175 is left in the supernatant after two weeks’ incubation. An AFM image of the sample shows straight rod structures (Figure 2a), which tend to align to each other. Using a transmission electron microscopy (TEM), we could observe more diverse structures (Figure 3) in the sample. There are rod structures of protein assemblies in gray, parallelly associated with each other (Figure 3a,b). We could also see images exhibiting darker needle-like structures, extending in length to hundreds of nanometers and bundling up in the perpendicular direction to its length (Figure 3b,c), which are the mineral precipitates. In Figure 3b, it is clearly seen that those mineral precipitates coexist with the gray protein rods. Sometimes, it may be difficult to distinguish them from each other. Comparing to TEM images for other conditions (TEM images for samples at other pH values shown in Appendix A), pH 6.0 is the only condition that a large amount of mineral precipitates could be observed, coexisting with protein aggregates. However, the exact identity of the mineral is still unknown.

An XRD 2θ profile of the precipitates at pH 6.0 (Figure 4A) was obtained and further compared to the XRD of crystals of octacalcium phosphate (OCP) (Ca_8_(HPO_4_)_2_(PO_4_)_4_∙5H_2_O), brushite (CaHPO_4_∙2H_2_O) and HAP [Ca_5_(PO_4_)_3_OH] (The Inorganic Crystal Structure Database OCD-65347-ICSD, brushite-16132-ICSD, HAP-169498-ICSD). The precipitates at pH 6.0 display an XRD profile with broader peaks than those of the crystals, however, the peaks match with those of HAP at most of the positions (Figure 4A), revealing the identity of the mineral formed at pH 6.0. The broader peaks suggest the non-homogenous nature of these precipitates compared to the crystal. In order to understand the function of amelogenin, a control experiment at pH 6.0 was set up using the same protocol, but without amelogenin. The experiment also resulted in a large amount of sand-like precipitates. However, the XRD profile of these precipitates after incubation at 37 °C for 4 weeks showed strong similarity to that of brushite, and no significant amount of HAP formed (Appendix A). Therefore, our results support that amelogenin is functional in converting the mineral calcium phosphate to a HAP crystalline structure at pH 6.0.

The XRD 2θ profiles of samples obtained at pH 4.5, 5.0 and 5.5 are also shown in Figure 4B for comparison, which are the same data as shown in Figure 1d–f. The protein amyloid diffraction peaks are clear in Figure 4B for pH 4.5, 5.0 and 5.5 at around 2θ = 19° and 9.0°, corresponding to 4.8 Å and 10.1 Å, respectively. However, such amyloid diffractions are not seen for pH 6.0 condition (Figure 4A). The XRD 2θ profiles showed brushite as the dominant mineral coexisting with the protein amyloid assembly at pH 5.0 and 5.5. There are peaks at 2θ positions around 28° and 41° in the XRD 2θ profiles that do not match the peaks for brushite, suggesting other minor mineral forms could coexist with brushite. The XRD profile of the sample prepared at pH 4.5 showed less similarity to the profile of all three crystals in Figure 4A, probably containing a mixture of minerals. 

Interestingly, for the samples prepared at pH 6.0 condition, different XRD results were obtained, which is consistent with our TEM studies. In Appendix A, the XRD of the sample prepared at pH 6.0 condition in the presence of H175 matches well with that shown in Figure 4. Different from Figure 4, the XRD shown in Figure 2b gives a better protein amyloid diffraction pattern with distances corresponding to 4.8 Å and 10.1 Å, however, little diffraction rings were observed for the mineral crystals. The samples were prepared using the same protocol at pH 6.0. The inconsistent results for the pH 6.0 samples were probably caused by the heterogeneous property of the samples, composed of both protein aggregates and mineral precipitates, as XRD only needs tiny amount of precipitates for the measurement, and the samples could contain different quantities of protein aggregates or mineral precipitates for different XRD runs.

### 2.3. ^31^P Solid-State NMR (SSNMR) Compared HAP Formed at pH 6.0 with the Natural Enamel

In order to know how comparable HAP formed at pH 6.0 to the natural human enamel, 1D ^31^P cross-polarization (CP) and 2D ^1^H-^31^P HETCOR experiments were carried out (pulse sequences shown in Appendix A). In Figure 5A, 1D ^31^P CP spectra of four samples were compared (preparation at pH 6.0 with and without amelogenin and preparation at pH 4.5 with amelogenin and human enamel). One sharp peak at 1.89 ppm is dominant for the sample at pH 4.5 with H175 and the blank sample at pH 6.0 without H175. A small peak at 0.27 ppm is only observed for the blank sample. Natural human enamel displays a broad peak at 3.54 ppm. The sample prepared at pH 6.0 in the presence of H175 shows a dominant peak at 3.54 ppm, the same position as that of the natural enamel. The results are consistent with the XRD observation that the conversion of mineral to HAP at pH 6.0 only occurs in the presence of H175. The 1D ^31^P CP spectrum at pH 6.0 with H175 shows a narrower line width than the spectrum of human enamel but contains a shoulder peak at around 1.89 ppm. 

Figure 5 also shows 2D ^1^H-^31^P HETCOR spectra from three samples (preparation at pH 6.0 with (B) and without (D) amelogenin, and human enamel (C)). The blank sample without H175 (Figure 5D) is different from the other two, showing a major crosspeak at (11 ppm/1.9 ppm) for (^1^H/ ^31^P), consistent with HPO_4_^2-^ (brushite) chemical shifts [24,25]. Minor crosspeaks at (0.5 ppm/3.54 ppm) and (5.5 ppm/3.54 ppm) could also be observed (seen in slice 1 only), but at a very low intensity. The 2D HETCOR spectra of HAP formed at pH 6.0 with H175 (Figure 5B) and human enamel (Figure 5C) display similar dominant peaks at (0.5 ppm/3.54 ppm) and (5.5 ppm/3.54 ppm). The crosspeak at (0.5 ppm/3.54 ppm) is from the correlation of OH^-^ group to ^31^P (PO_4_^3^^−^) for HAP (Ca_5_(PO_4_)_3_OH) and the crosspeak at (5.5 ppm/3.54 ppm) originates from the adsorbed H_2_O correlation to ^31^P atoms (PO_4_^3-^) [26]. The spectrum is the same as that reported in the literature on enamel or HAP in an aqueous environment [26]. This shows the protocol used in this report at pH 6.0 in the presence of H175 is capable of converting calcium mineral into HAP, mimicking natural enamel. Comparing slice 1 (^31^P at 3.54 ppm) of Figure 5B,C, the relative intensity of peaks correlating to the OH^-^ group and adsorbed H_2_O is very different. The correlation peaks of the OH^-^ group are stronger than those of adsorbed H_2_O, since the OH^-^ group is part of the crystal structure, closer to the PO_4_^3-^ group than adsorbed water bound on the surface of the crystal. For the natural enamel sample, there is no peak at the ^1^H 5.5 ppm position if water is not added. However, even though additional water was added to the powdered natural enamel sample, the peak of bound H_2_O is still relatively small, suggesting more compactness and less surface area of the natural enamel sample. Comparing slice b of Figure 5B,C (^1^H at 5.5 ppm), the peak in slice b of Figure 5B exhibits a narrower linewidth but a shoulder at around 1.9 ppm, consistent with the 1D ^31^P CP result. The existence of a shoulder peak at 1.9 ppm for the sample prepared at pH 6.0 suggests that the conversion of mineral to HAP using amelogenin at pH 6.0 is not 100%; there is a small amount of other mineral components. 

### 2.4. Amyloid-like Assemblies of H175 are Facilitated by a Short Segment of Sequence

SSNMR was also used to elucidate the structure of H175 assemblies formed at pH 6.0. The same sample for ^31^P SSNMR study of minerals was used in the protein structural investigation. The 2D CP based ^13^C-^13^C correlation spectrum with DARR mixing is shown in Figure 6A. The spectrum exhibits a small number of peaks considering H175 with 175 residues in the sequence. The peak linewidth is about 0.7 ppm for some well-resolved peaks. However, it is difficult to assign the peaks since multi-dimensional NMR experiments, such as NCaCx, exhibit very weak signals (Figure 6B). The signal intensity is limited by the small quantity of protein in the rotor, where most of the rotor volume is taken by the mineral precipitation (about 10 mg of protein and a total sample weight of about 67 mg).

We then obtained an SSNMR 2D ^13^C-^13^C correlation spectrum for the protein aggregates formed at pH 4.5 for comparison. This sample had more protein and less mineral precipitation, since calcium phosphate salt has higher solubility at this acidic condition. To our surprise, the 2D ^13^C spectrum is almost the same as that for pH 6.0 (Figure 6C, overlay spectra in Appendix A), suggesting similar secondary structures shared by these two different pH conditions. Multi-dimensional NMR experiments including 2D NCaCx (Figure 6D), 2D NCoCx (Figure 6E), 3D NCaCx and 3D NCoCx (Appendix A) were carried out for the sequential assignment. The assignment is shown in Appendix A, for segment ^18^EVLTPLKWYQSI^29^. ^14^NFSY^17^ and ^30^RP^31^ were also tentatively assigned based on residue type and proximity to the assigned sequence ^18^EVLTPLKWYQSI^29^. The chemical shift index (Figure 7A) and TALOS-N prediction [27] (Figure 7B) indicate β-strand conformation for residue 19-30. Considering the close similarity in the spectra obtained for protein assemblies at pH 6.0 and 4.5, the assignment of the ^13^C-^13^C spectrum in Figure 6A was therefore copied from Figure 6C directly. Our SSNMR results strongly support a segmental β-sheet structure for protein assemblies formed at pH 6.0, confirming the XRD results.

Besides the assigned β-sheet segment, there are some peaks clearly shown in both ^13^C-^13^C spectra (labeled in Figure 6A, C as An, In1, In2, Ln1, Ln2, Pn1, Pn2, Sn, Tn1, Vn1, Vn2). These peaks could only be labeled according to the residue type. This strongly supports that other rigid segments exist besides the β-strand region ^18^EVLTPLKWYQSI^29^. The chemical shift values of these peaks are listed in Appendix A. The secondary chemical shifts were also calculated and showed mostly negative values or small absolute values close to 0, suggesting those residues could adopt β-sheet or random coil secondary structures.

### 2.5. H175 has Other Structural Components beside β-amyloid

From the cross-polarization (CP)-based ^13^C-^13^C correlation spectrum, it can be concluded that the rigid segments in H175 assemblies at pH 6.0 contain more than the identified β-sheet sequence. However, many residues are still missing in the SSNMR spectra. In order to find the dynamic segment of H175 assemblies at pH 6.0 condition, insensitive nuclei enhanced by polarization transfer (INEPT)-based 2D total through bond correlation spectroscopy (TOBSY) was carried out, showing a small number of peaks too (Figure 8A). The assignment based on the residue type is listed in Appendix A for INEPT-TOBSY. Judging from the secondary chemical shifts value, those residues could also adopt β-sheet or random coil secondary structures. Considering only a limited number of residues are observed using both CP-based and INEPT-based experiments, these results indicate a large number of residues in H175 assemblies are in disorder, or in motion with the intermediate timescale, invisible for NMR. 

A further comparison between the CP-based ^13^C-^13^C correlation spectrum (Figure 6A) and INEPT- based TOBSY spectrum (Figure 8A) is shown in Figure 8B. To our surprise, there are several peaks shown in both spectra. These peaks with the matched chemical shifts within ±0.4 ppm include the crosspeaks for one I (In1/I2), two L residues (Ln1/L1) (Ln2/L2), two V residues (Vn1/V1) (Vn2/V2), two P residues (Pn1/P2) (Pn2/P1), one S (Sn/S1) and one T (Tn1/T1). It is unclear whether those peaks seen in both spectra represent the same residues in the sequence. There are also several E/Q residues (E/Q1, E/Q2, E/Q3 in TOBSY) that probably also exist in both spectra, for example, E/Q2 in TOBSY is very close to E18 in the CP-based ^13^C-^13^C correlation spectrum. However, most of these residues are not located in the sequentially assigned β-sheet region. If these overlapped peaks represent the same residues in both spectra, it suggests these residues could be in two states, dynamic and rigid. The average conformation of these dynamic residues could be the same as these residues in rigid state upon the conformation becoming fixed. The results suggest that although a small part of H175 could adopt β-sheet conformation and be very rigid, the protein also has other structural components with different dynamics. 

INEPT-based 2D TOBSY was also carried out for the sample prepared at pH 4.5 condition for comparison (Appendix A). By comparing the spectra of samples from pH 4.5 and pH 6.0, we found the two spectra were similar only at some positions. Several peaks disappear for the pH 4.5 condition (mostly Cα-Cβ crosspeaks, Appendix A). The results indicate that although the two samples have the same β-sheet segment and other rigid components, some other segments of protein could still be different in the dynamics.

## 3. Discussion

Essential conditions for HAP formation 

Our experiments were carried out in the presence of both calcium ions and phosphate ions. Without the protein, the calcium phosphate mineral would start to precipitate at a slight acidic condition, since Ca(H_2_PO_4_)_2_ is highly soluble and CaHPO_4_ is not soluble with Ksp = 10^−6.90^ [28]. The Ka_2_ of phosphoric acid is 10^−7.21^. At pH 6.0, there would be coexistence of both ions of [H(PO_4_)^2-^] and [H_2_(PO_4_)^-^]. We observed a large amount of mineral precipitates formed at pH 6.0 and proved the main component is brushite (CaHPO_4_∙2H_2_O) using XRD and ^31^P NMR. This is consistent with the chemical equilibrium prediction, and brushite could not be converted into other calcium phosphate forms after 4 weeks’ incubation. However, in the presence of H175, the mineral would be converted into HAP [Ca_5_(PO_4_)_3_OH], which has much lower solubility with Ksp = 10^−58.4^ [28], and so far, how amelogenin controls the mineral form is not fully understood. 

The pH is also another important factor in determining the mineral precipitation. Although our results indicate that the amyloid formation could happen in a big pH range (pH 4.5–6.0), the precipitated mineral components could be very different. HAP could only form at pH 6.0. It has been observed in many laboratories that amelogenin is functional in vitro in inducing HAP formation. Most of the conditions used in the literature involving amelogenin are at near neutral pH (pH 7.4-7.6) with protein concentration at sub-milligram/mL. This pH is just past pKa2 (7.21) of phosphoric acid, therefore, more mineral precipitation would form. Our results clearly indicate that the pH value is not necessarily so close to neutral. At pH 6.0, a slightly acid condition, HAP could still form with the help of amelogenin. From NMR, the secondary structures for the rigid segment of H175 assemblies are the same at pH 4.5 and 6.0. 

Could we draw the conclusion that amyloid-like assemblies are functional in promoting HAP formation? Although previous SSNMR studies have shown that amelogenin at functional conditions have β-sheet secondary structural components at some sites, the 2D ^13^C spectra of the uniformly ^13^C-labeled sample gave less information because of the low resolution and sensitivity [17]. With a higher protein concentration (1–2 mg/mL), our experimental condition made it easier to collect the large amount of protein assemblies needed for NMR study. Utilizing SSNMR, we were able to study the structure of the mineral and the protein assembly at the same time using the same sample, providing an opportunity to study the functional state of the protein and its relationship with the minerals. Our SSNMR spectrum on uniformly ^13^C-labeled H175 shows a better resolution, indicating multiple structure components including a β-sheet segment. However, the role of an amyloid-like structure in determining HAP formation is still not conclusive. All the structural elements are important and play a role together in controlling HAP formation. Furthermore, the NMR signal is dominant for the bigger population of the sample; it is still possible that two or more populations of protein assemblies exist in the sample at pH 6.0. One population is dominant, containing the amyloid-like structure, while the other minor population with an unknown structure may play the real regulating role. A further investigation on amelogenin at other functional conditions is also needed to confirm the necessity of the amyloid-like structure in promoting HAP formation. Peptides derived from amelogenin are functional in enamel biomimetics. It would be interesting to investigate the structure of those peptides. From previous research, we have confirmed that LRAP is able to form a similar amyloid-like structure, supporting the importance of an amyloid-like structure [21]. 

## 4. Materials and Methods

### 4.1. Recombinant Amelogenin Protein Expression and Purification

The full-length human amelogenin gene (AMELX) sequence (ID:M86932.1 in GenBank) was synthesized and inserted into a vector (General Biosystems Co. Ltd., Anhui, China). The gene was further cloned to pET3a SUMO plasmid with a N-terminal 6×His tag and transformed into *E. coli* Rosetta (DE3) cells for expression. The recombinant protein was expressed in LB medium and induced at 37 °C using 0.8 mM isopropyl β-D-thiogalactoside at OD_600_ of ~ 0.8, for 4 h. The cells were harvested using centrifugation. The cells were then resuspended in a lysis buffer (50 mM Tris-HCl, 300 mM NaCl, pH8.0) and lysed using a high pressure nano homogenizer (FB-110X, Shanghai Litu Ins., Shanghai, China). The supernatant of the lysis after centrifugation was incubated with Ni-NTA beads for 2 h. Subsequently, TEV (tobacco etch virus) protease was added to the eluent from the Ni column to remove the SUMO fusion tag. After the enzyme cleavage, an extra G was left at the N terminus of the amelogenin protein sequence (H175). The protein was then dialyzed against deionized water and lyophilized. For further purification, the protein was dissolved in 2% acetic acid and purified using C4 hydrophobic chromatography (Bio-C4, Sepax Technologies Ins., Suzhou, China).

The uniformly [^13^C, ^15^N]-labeled protein was first grown in M9 medium containing 4 g/L ^12^C-glucose and 1.5 g/L ^14^N-ammonium chloride. At OD_600_ ≈ 1.0, the cell pellets were transferred to 1 L M9 medium containing 2 g/L ^13^C-glucose and 1.5 g/L ^15^N-ammonium chloride. The further expression and purification method are the same as above.

### 4.2. Amelogenin Protein H175 Self-assembly Preparation

Both ions and pH have a tremendous effect on amelogenin self-assembly. H175 was first dissolved in 0.001N HCl and diluted to an aqueous solution containing 33.4 mM of CaCl_2_ and 20.9 mM of KH_2_PO_4_ to a final concentration of 1mg/mL or 2 mg/mL. The solution pH was varied from 4.5, 5.0, 5.5 to 6.0 by adding 0.1M KOH. At a specific pH, the solution was incubated at 37 °C for more than 7 days for protein self-assembly and mineral precipitation. 

The uniformly [^13^C, ^15^N]-labeled H175 self-assembly samples were made using 16 mL 2 mg/mL protein solution at the pH 4.5 condition and 20 mL 1mg/mL protein solution at pH 6.0 condition. The samples were incubated at 37 °C for a month to ensure the maximum protein conversion into the precipitates and to allow time for the mineral structural conversion. After that, the sample was collected using centrifugation at 259,000 g, 20 °C for 1 h (Optima Max-TL, BECKMAN COULTER, Bera, CA, USA). A blank sample was also prepared using the same experimental condition but without H175.

### 4.3. Enamel Isolation from Dental Tissues

Human enamel was isolated from the dental tissues of ca. 4 healthy molars. The enamel was first powdered using high speed grinding provided by a dental comprehensive treatment machine. The sample was then dried in air. Finally, the dried sample was further ground into more homogeneous powder with a mortar.

### 4.4. X-Ray Diffraction (XRD)

All samples were collected using centrifugation at 259,000 g, 20 °C for 1 h (Optima Max-TL, BECKMAN COULTER, Bera, CA, USA). The sample composed of protein aggregates and salt crystals in nanometer size was mounted on a loop. The XRD measurements were carried out using a single-crystal X-ray diffractometer (D8 VENTURE, Bruker, Karlsruhe, Germany) with Cu Kα radiation (λ = 0.154184 nm) operating at an acceleration voltage of 50 kV and a current of 1 mA. The distance from crystal to detector was 50 mm with variable exposure time 20–60 s.

The sample diffraction was recorded at two temperatures, either 298 K or 150 K. For all samples prepared at pH 4.5, pH 5.0 and pH 5.5 conditions and some samples prepared at pH 6.0, the diffraction was measured at 298K, while for one sample prepared at pH 6.0 and the blank sample without protein (shown in Appendix A), the diffraction was recorded at 150K to get better diffraction. The XRD images were processed with the adxv.x86_64RHEL6 program (Scripps Research Institute, La Jolla, CA, USA). Then, the data were radially integrated and further processed with an X’Pert HighScore Plus (PANalytical B. V., Almelo, Netherlands) to be transformed into the powder pattern. The XRD of HAP, OCP and brushite were obtained from the Inorganic Crystal Structures Database (ICSD, FIZ Karlsruhe GmbH, Germany).

### 4.5. Atomic Force Microscopy (AFM)

Aliquots (20 μL) of solution at different incubation periods were pipetted onto a freshly cleaved mica surface and kept for 30 min in a wet cell to avoid evaporation. Subsequently, the mica surface was rinsed twice with 20 µL Milli-Q water by first keeping the water on the mica surface for 2 min and then removal of the water using a rubber suction bulb. AFM was carried out under the dry condition in the tapping mode using Si cantilevers with a tip radius of 8–12 nm and 40 N/m force constant (model RTESPA-300, Bruker, Camarillo, CA, USA) at about 300 kHz on a Dimension ICon AFM with a Bruker Nanoscope V controller (Digital Instruments, Goleta, CA, USA). 

### 4.6. Transmission Electron Microscopy (TEM)

5 µL of solution incubated for more than 7 days was taken out periodically and was analyzed using TEM. The solution was first adsorbed onto a carbon-coated copper grid (300 meshes, Beijing Zhongjingkeyi Technology Co., Ltd., Beijing, China) for 1 min, then blotted away using filter paper. Immediately after, the grid was washed with three drops of Milli-Q water. For negative staining, the grid was stained with 5 µL of 2% uranyl acetate in water (*w*/*v*) for 10 s and dried quickly with filter paper. To get a better staining effect, the grid was stained using another drop of 5 μL 2% uranium acetate for 1 min and washed off with one drop of water again. Finally, the grid was air dried. The TEM images were recorded on a Tecnai G2 Spirit Transmission Electron Microscope with an acceleration voltage of 120 keV. 

### 4.7. Magic Angle Spinning (MAS) Solid-State NMR Experiments

All NMR experiments were acquired on a 16.4 T (^1^H frequency, 700 MHz) Bruker AVANCE NEO spectrometer. ^13^C-^13^C and ^13^C-^15^N correlation spectra were collected with a 3.2 mm triple-resonance HCN Bruker probe under 15 kHz MAS, while a 3.2 mm HCP MAS probe was utilized to perform all ^1^H-^31^P experiments under the same MAS rates. ^13^C chemical shifts were externally referenced to DSS by setting a downfield ^13^C signal of adamantine to 40.48 ppm, and ^15^N chemical shifts were indirectly referenced to liquid ammonia (0 ppm). ^31^P chemical shifts were indirectly referenced to 85% H_3_PO_4_ (0 ppm) using the resonance frequency ratio 0.404807420. All the experiments were collected with a 2 s recycle delay.

2D ^13^C-^13^C DARR [29,30], INEPT-TOBSY [31,32], ^15^N-^13^C NCa, NCaCX, NCo, NCoCx and 3D NCaCx and NCoCx [33,34,35,36,37] spectra were acquired to get resonance assignments with the samples prepared at pH 4.5 and pH 6.0 conditions. The sample temperature was set to 273 K for all the MAS experiments, except the 2D DARR and INEPT-TOBSY experiments for the sample at pH 4.5, where the temperature was set to 293 K. For both the samples at pH 4.5 and pH 6.0, the band-selective transfers [38] of NCa and NCo experiments were implemented with a contact time of 4 ~ 5.5 ms, while ^15^N (centered at 120 ppm) rf-field strengths were optimized to 5/2 ωr (37.5 kHz), and ^13^Ca (centered at 53 ppm) and ^13^Co (centered at 173 ppm) rf-field strengths were optimized to 3/2 ωr (22.5 kHz) and 7/2 ωr (52.5 kHz), respectively. 2D DARR spectra were recorded using a CP contact time of 1.8 ms and ^1^H fields of 79.77 kHz. The mixing time was varied from 25 to 500 ms. During the acquisition, SPINAL-64 [39] ^1^H decoupling was applied with the ^1^H field strength of 83.3 kHz. The 2D ^13^C-^13^C INEPT-TOBSY experiment was recorded with a TOBSY mixing time of 11.2 ms using the P9^1^_6_ mixing sequence.

1D ^1^H-^31^P CP and 2D ^1^H-^31^P HETCOR [40] experiments were also performed for all the four samples, which were the natural human enamels, H175 assemblies incubated at pH 4.5, H175 assemblies incubated at pH 6.0 and the blank sample incubated at pH 6.0 without proteins. All the measurements were done at 288 K. The CP contact time was 1.5 ms with ^1^H fields of 75.85 kHz, and the ^1^H 90° pulse was 3.2 µs. During the acquisition, SPINAL-64 [38] ^1^H decoupling was also applied with a ^1^H field strength of 83.3 kHz.

All 2D spectra were processed using TopSpin 4.0.2 (Bruker Biospin, Rheinstetten, Germany) and apodized in all dimensions with a shifted sine-bell window function (SSB = 2). Subsequently, the spectra were analyzed in detail using Sparky.

## Figures and Tables

**Figure 1 ijms-21-02946-f001:**
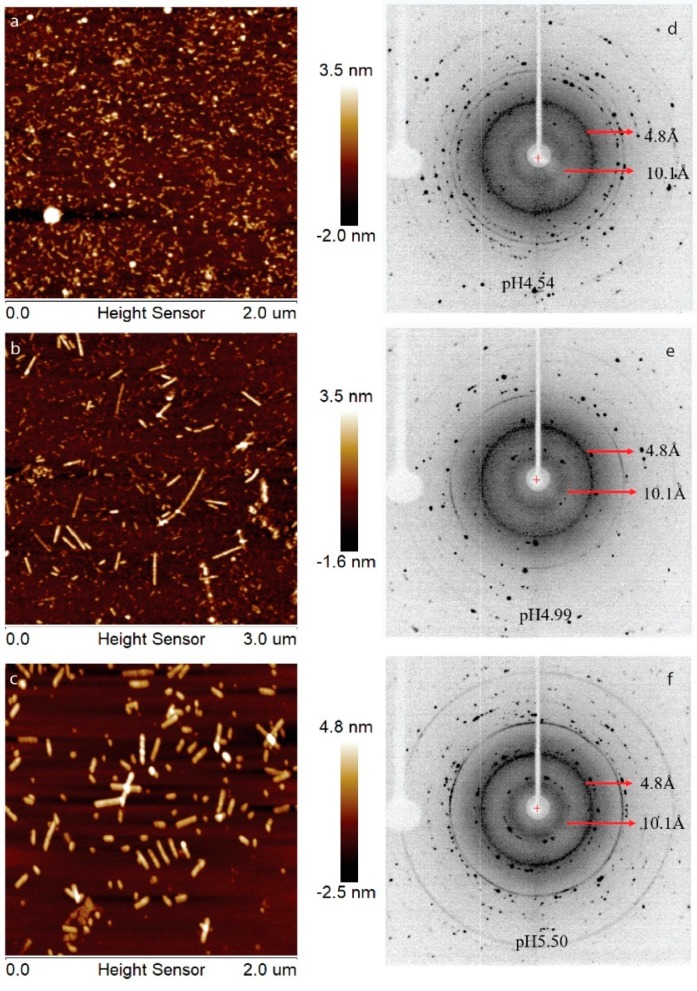
AFM images, (**a**–**c**), and XRD images, (**d**–**f**), of H175 assemblies prepared at pH 4.5, 5.0 and 5.5 at 37 °C. (**a**,**d**) H175 assemblies at pH 4.5. The incubation period is 20 days. (**b**,**e**) H175 assemblies at pH 5.0. The incubation period is 20 days. (**c**,**f**) H175 assemblies at pH 5.5. The incubation period is 14 days. The arrows in XRD indicate the equatorial and meridional reflections at 10.1 Å and 4.8 Å, respectively. The other Debye rings in XRD indicate salt crystal reflection.

**Figure 2 ijms-21-02946-f002:**
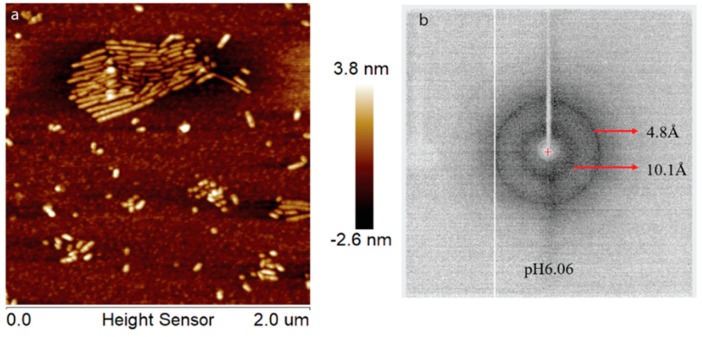
AFM image (**a**) and XRD image (**b**) of H175 assemblies formed at pH 6.0 after 14 days’ incubation at 37 °C. The arrows in (**b**) indicate the equatorial and meridional reflections at 10.1 Å and 4.8 Å, respectively.

**Figure 3 ijms-21-02946-f003:**
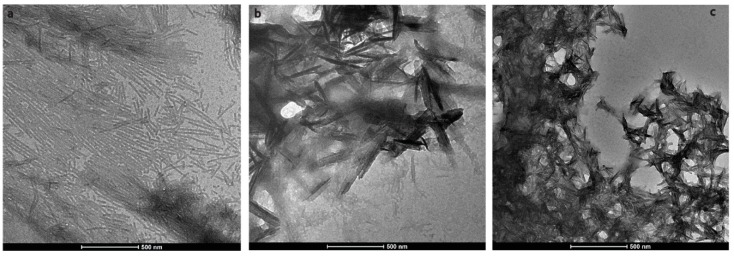
Transmission electron microscopy (TEM) images of H175 self-assembly at pH 6.0 after 14 days’ incubation at 37 °C. (**a**) shows the images of the protein assemblies in rod structures parallelly associated with each other. (**c**) displays another view dominant with the darker needle-like structure. (**b**) shows longer needle-like structures with the gray protein rod structures coexisting.

**Figure 4 ijms-21-02946-f004:**
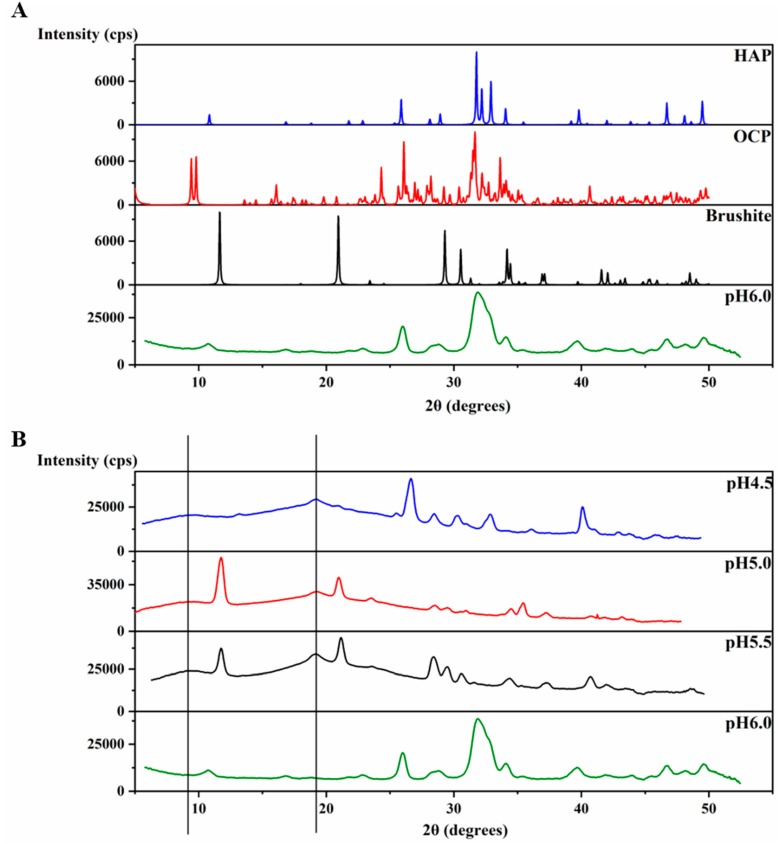
Characteristic XRD patterns for H175 self-assembly at different pHs. (**A**) Comparison of the X-ray powder diffractogram. The sample formed at pH 6.0 is the mixture of the H175 assembly and mineral precipitation. XRD of crystals of octacalcium phosphate (OCP), brushite and hydroxyapatite (HAP) were downloaded from the Inorganic Crystal Structure Database (OCD-65347-ICSD, brushite-16132-ICSD, HAP-169498-ICSD). (**B**) Comparison of the X-ray powder diffractogram of the samples prepared at pH 4.5, pH 5.0, pH 5.5 and pH 6.0. (The black lines represent the d-spacing of 4.8 Å and 10.1 Å for protein assemblies.).

**Figure 5 ijms-21-02946-f005:**
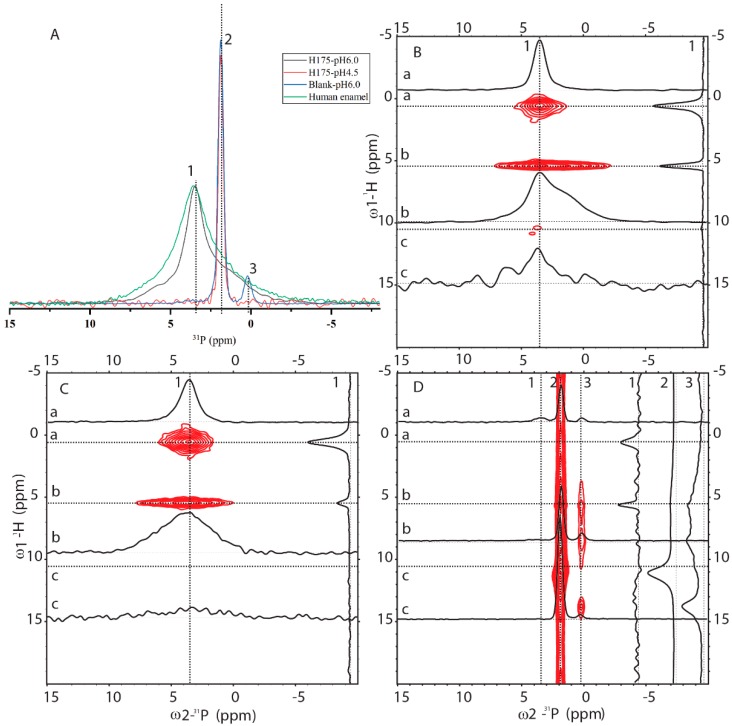
Solid-state NMR (SSNMR) spectra of minerals prepared at pH 6.0 (with H175), pH 4.5 (with H175), blank (a control sample prepared under the same solution condition as the sample at pH 6.0 but without H175) and a natural human enamel sample (number of scans = 64). Line 1, 2 and 3 indicate 3 positions at δp = 3.54 ppm, δp = 1.89 ppm and δp = 0.27 ppm. (**A**) 1D ^1^H-^31^P CP-MAS spectra. The intensity of the spectra was normalized for easy comparison of the peak positions. The 2D ^1^H-^31^P HETCOR spectra acquired from the samples prepared at pH 6.0 with (**B**) or without (**D**) H175, and the natural human enamel sample (**C**). In HETCOR spectra, dotted lines a, b, c show 3 positions at ^1^H dimension, corresponding to 0.5 ppm, 5.5 ppm and 11 ppm, respectively. 1D slices are labeled accordingly.

**Figure 6 ijms-21-02946-f006:**
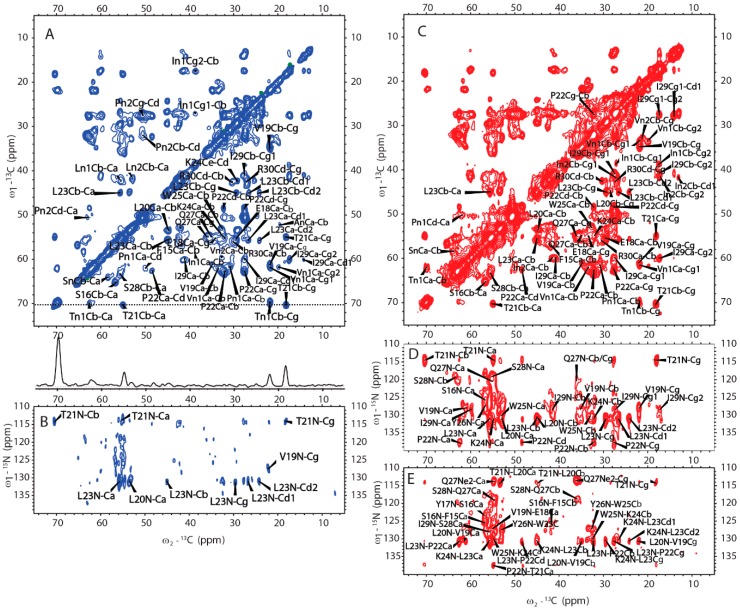
The MAS SSNMR spectra acquired from the samples prepared at pH 6.0 (left) and at pH 4.5 (right) using uniformly [^13^C, ^15^N]-labeled H175. (**A**) 2D ^13^C-^13^C DARR correlation spectrum with a DARR mixing time of 50 ms and (**B**) 2D ^13^C-^15^N NCaCx spectrum for the sample prepared at pH 6.0. (**C**) 2D ^13^C-^13^C DARR correlation spectrum with a DARR mixing time of 25 ms. (**D**) 2D ^13^C-^15^N NCaCx spectrum. (**E**) 2D ^13^C-^15^N NCoCx spectrum for the sample prepared at pH 4.5.

**Figure 7 ijms-21-02946-f007:**
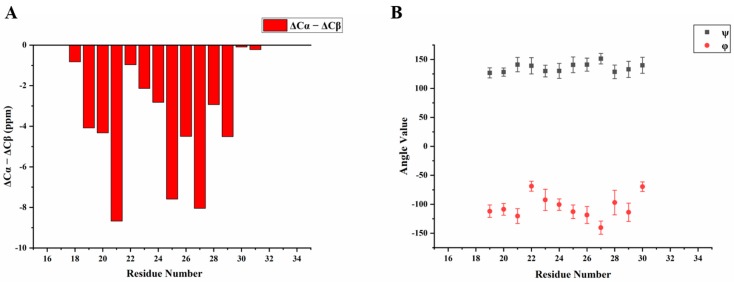
Secondary structure prediction based on the chemical shifts assignment of H175 self-assembly at pH 6.0. The sequence of H175 is displayed on the top, highlighting the sequentially assigned sequence in red (^18^E - ^31^P). (**A**) The calculated secondary chemical shifts (ΔCα-ΔCβ) show the negative values for the identified sequence. (**B**) Predicted protein dihedral angles ψ and φ using TALOS-N, indicating β-sheet secondary structures.

**Figure 8 ijms-21-02946-f008:**
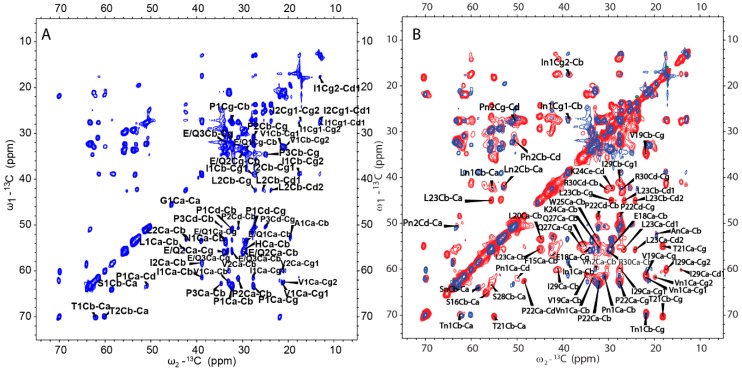
(**A**) 2D ^13^C-^13^C INEPT-TOBSY spectrum of H175 assemblies formed at pH 6.0 using uniformly [^13^C, ^15^N]-labeled protein. (**B**) Comparison between 2D DARR spectrum (red) and 2D INEPT-TOBSY spectrum (blue) of H175 assemblies prepared at pH 6.0.

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
