# Peer review of "Hydroxyapatite Formation Coexists with Amyloid-like Self-Assembly of Human Amelogenin"

_ijms, 2020, doi:10.3390/ijms21082946_

Round 1

Reviewer 1 Report

The manuscript entitled “Hydroxyapatite formation coexists with amyloid-like self-assembly pf human amelogenin” by Jiang Zhang et. al. described a critical in vitro condition for the formation of HAP in the existence of full-length human amelogenin (H175). The screening of the acidic conditions was fully characterized using multiple techniques such as AFM, XRD, TEM, and SSNMR. The aim of the research is carefully designed, the methods used are well documented, results are convincing and the conclusion is mostly supported by the experimental evidence. The manuscript provides the high-resolution segmental structures of H175 via multi-dimensional SSNMR methods, resulting from tremendous amount of careful experimental designs and work. The determined segmental structures can be interesting to the readers in the field. The manuscript is well prepared and I recommend to publish this work on Int. J. Mol. Sci. with following minor suggestions:  

Page 2, line 68: change the phrase of “in XRD” and “in 31P 1D and 2D …” to “by XRD” and “by 31P 1D and 2D …”

Page 6, line 191 to line 192: a broad peak centered around 1.9 ppm corresponding to brushite is mostly absence in human enamel and the sample prepared at pH 6. It would be appreciated if the authors can provide a reasonable explanation for the difference.  

Page 7, Figures and other Figures in the manuscript: font size of the axis and axis labels of the NMR spectra should be large enough to be seen.

Page 7, line 214: Change a word of “little” to “less”.

Page 8, Figure 6: First of all, please edit the Figure caption so that to reflect the fact that all spectra on the left column are related to the sample prepared at pH of 4.5.  It seemed to me that when authors compared two spectra acquired for the samples prepared at pH6 and pH 4.5. they neglected fact that those two spectra were acquired at different mixing times (50 ms. and 50 ms.).  Why the authors decided to compare the spectra acquired with different mixing times?

Page 8, line 225: I think your data strongly suggested a segmental b-sheet, instead of the whole b-sheet for the assemblies.  

Page 9, line 242: change a word “more” to “many”.

Reviewer 2 Report

The paper entitled Hydroxyapatite Formation Coexists with 2 Amyloid-like Self-assembly of Human Amelogenin presents a solid state structural studies on enamel formation.

The paper reports X-ray diffraction as well as 31P Solid state NMR studies. The paper posses some elements of novelty, but unfortunately probably approaching the lower limit of what is considered a significant and value-added crystallographic discussion. Although the data could be of interest to some readers, generally, throughout the crystallographic part of paper the proper presentation of it is missing. I can recommend publication of this paper in the International Journal of Molecular Science only after major revisions.

Despite I belive it was difficult to obtain good quality crystallographic data I think that the crystallographic work should be better performed:

  • In 4. Materials and Methods data related to XRD measurements are missing. No necessary information is displayed.

-Please add experimental data related to X-ray diffraction.

Scarce information was presented:

,,The sample collected was studied using a single-crystal X-ray Diffractometer (Bruker D8 359 VENTURE, Cu Kα, λ = 0.154184 nm) operating at an acceleration voltage of 50 kV and a current of 1 360 mA at either 298 K or 150 K.''

-Why did you performe measurement at room temperature and 150 K?

-What about powder X-ray diffraction measurement? Generally, more details through experimental study in PXRD analysis should be reported; please report: diffractometer type, used geometry, a scanning speed, radiation type, programs used for data collection and visualization,

  • In 2. Results: 2.2. HAP is formed at pH 6.0 in the presence of H175 amyloid

-You have stated:,,The broader peaks suggest the non-homogenous nature of these precipitates compared to the crystal. '' Is that so? Please explain sample preparation. What about crystallite size?

-Are you sure that the XRD peaks showed brushite as a dominant mineral coexisting with the protein amyloid assembly at pH 5.0 and pH 5.5? What about peak at positions around 2Q 28 and 41 degree?

Round 2

Reviewer 2 Report

Recommendation: Publish without change.